# Decrease in the prevalence of hepatitis B and D virus infections in an endemic area in Peru 23 years after the introduction of the first pilot vaccination program against hepatitis B

Cesar Cabezas[1,2]*, Omar Trujillo[3], Johanna Balbuena📷[1], Flor de Maria Peceros[1], Manuel Terrazas[1], Magna Suárez[1], Luis Marin[1], Janet Apac[4], Max Carlos Ramírez-Soto📷[1]

1 Centro Nacional de Salud Puública, Instituto Nacional de Salud, Lima, Peru, 2 Facultad de Medicina, Universidad Nacional Mayor de San Marcos, Lima, Peru, 3 Centro Nacional de Salud Intercultural, Instituto Nacional de Salud, Lima, Peru, 4 Dirección de Epidemiología, Dirección Regional de Salud de Apurímac, Apurímac, Peru

* ccabezas@ins.gob.pe

## Abstract

In 1991, Peru launched the first vaccination program against hepatitis B in children aged under 5 years in the hyperendemic [hepatitis B virus (HBV) and hepatitis D virus (HDV)] province of Abancay. We conducted a cross-sectional study to determine the prevalence of HBV and HDV infections, 23 years after the launch of the vaccination program, as well as the post-vaccine response against hepatitis B in terms of prevalence of hepatitis B surface antibody (anti-HBs ≥10 mUI/ml). Among 3165 participants aged from 0 to 94 years, the prevalence rates of hepatitis B surface antigen (HBsAg), and hepatitis B core antibody (total anti-HBc) were 1.2% [95% confidence interval (CI) 0.85–1.64%], and 41.67% (95% CI 39.95–43.41%), respectively. The prevalence rate of anti-HBs at protective levels (≥10 mUI/ml) in individuals who HBsAg and anti-HBc negative was 66.36% (95% CI 64.15–68.51%). The prevalence rate of HBsAg in children aged <15 years was nil, and among adult HBsAg carriers, the prevalence of hepatitis D antibody (anti-HDV) was 5.26% (2/38; 95% CI 0.64–17.74). These findings showed that HBV prevalence has changed from high to low endemicity, 23 years following implementation of the vaccination program against hepatitis B, and HDV infection was not detected in those aged <30 years.

## Introduction

Hepatitis B virus (HBV) infection remains a serious public health problem worldwide [1]. Since 1984, prevention programs against HBV infection have been implemented in various countries, including immunization using hepatitis B vaccine and use of immunoglobulin in infants born to hepatitis B surface antigen (HBsAg)-positive mothers [2,3], which have led to a substantial decline in the prevalence of HBV and hepatitis D virus (HDV) infections among children in hyperendemic countries such as China, Japan and Colombia [2–6]. In this context,

**Data Availability Statement:** All relevant data are within the manuscript and its Supporting Information files.

**Funding:** This study was supported by the Instituto Nacional de Salud, Lima -Peru [project OI- 083-2013].

**Competing interests:** The authors have declared that no competing interests exist.

a model study has shown that vaccination of infants and neonates is already driving a significant decrease in new infections; vaccination has already prevented 210 million new cases of chronic infections by 2015 and will have averted 1.1 million deaths by 2030 [7]. Similarly, morbi-mortality for HBV-related liver diseases has substantially declined in children and adults since the introduction of vaccination against HBV [8,9].

In 1991, Peru launched the first vaccination program against HBV in children aged under 5 years in the HBV- and HDV-hyperendemic province of Abancay [10]. The program began with immunization of 3791 children aged under 5 years between 1991 and 1994, and subsequently expanded to universal vaccination in Abancay as well as other hyperendemic provinces in Peru [10]. Following implementation of the program against HBV, the carrier rate of HBsAg decreased from 9.8% in 1991 (in the general Abancay population) [11] to below 3% in 2010 among those aged ≥18 years (including 2.5% and 1.9% of university students and blood donors, respectively) [12,13]. Over the next 22 years, this vaccination program has also led to a decline in the mortality burden of HBV-related liver diseases, particularly cirrhosis and fulminant hepatitis, in children aged under 15 years [14].

Despite the benefits of the vaccination program in terms of lower carrier rates of HBsAg in those aged ≥18 years and reduced mortality rates from HBV-related liver diseases [12–14], to date, there are no data on the impact of the HBV vaccination program on the carrier rates of HBsAg, hepatitis B core antibody (anti-HBc), hepatitis B surface antibody (anti-HBs) and hepatitis D antibody (anti-HDV) in the general population of Abancay. Therefore, in this study, we determined the prevalence of HBV and HDV infections in Abancay province 23 years since the introduction of the first pilot vaccination program against HBV, as well as the post-vaccine response against hepatitis B in terms of prevalence of anti-HBs. These findings would prove useful in determining the effectiveness of the program, in order to mitigate the mortality risk and strengthen vaccination coverages in at-risk populations, with an overall aim to eradicate HBV infection in Abancay by 2030.

## Material and methods

### Study design

A cross-sectional study was conducted in the general population of the province of Abancay in Peru between November and December 2014.

### Study population and sample method

The province of Abancay is the departmental capital of Apurímac, a poor area in the south central highlands of Peru. Abancay has nine districts, each with a population ranging between 1213 and 51,225 inhabitants. The overall estimated population of Abancay in 2007 was 96,064, according to a regularly updated census of the National Institute of Statistics and Information (INEI) (Fig 1).

The required sample size was calculated using the design effect. Based on 95% confidence interval (CI), a margin of error of 4.3%, a design effect of 2, a prevalence rate of 50%, and a response rate of 80%, the required sample size was 3520 participants.

Conglomerate (population groups) stratified random sampling and multi-stage were used in this study. First, conglomerates in each geographical area were selected using INEI's 2007 national population census and housing data. A total of 113 conglomerates comprising 32 inhabitants each were selected for analysis. Second, conglomerate houses were selected, based on urban and rural population lists derived from the census. In cases of rejections or losses, the houses were replaced by the next eligible house on the list. Third, participants were selected, based on the birthday date closest to the date of the survey visit; otherwise, the

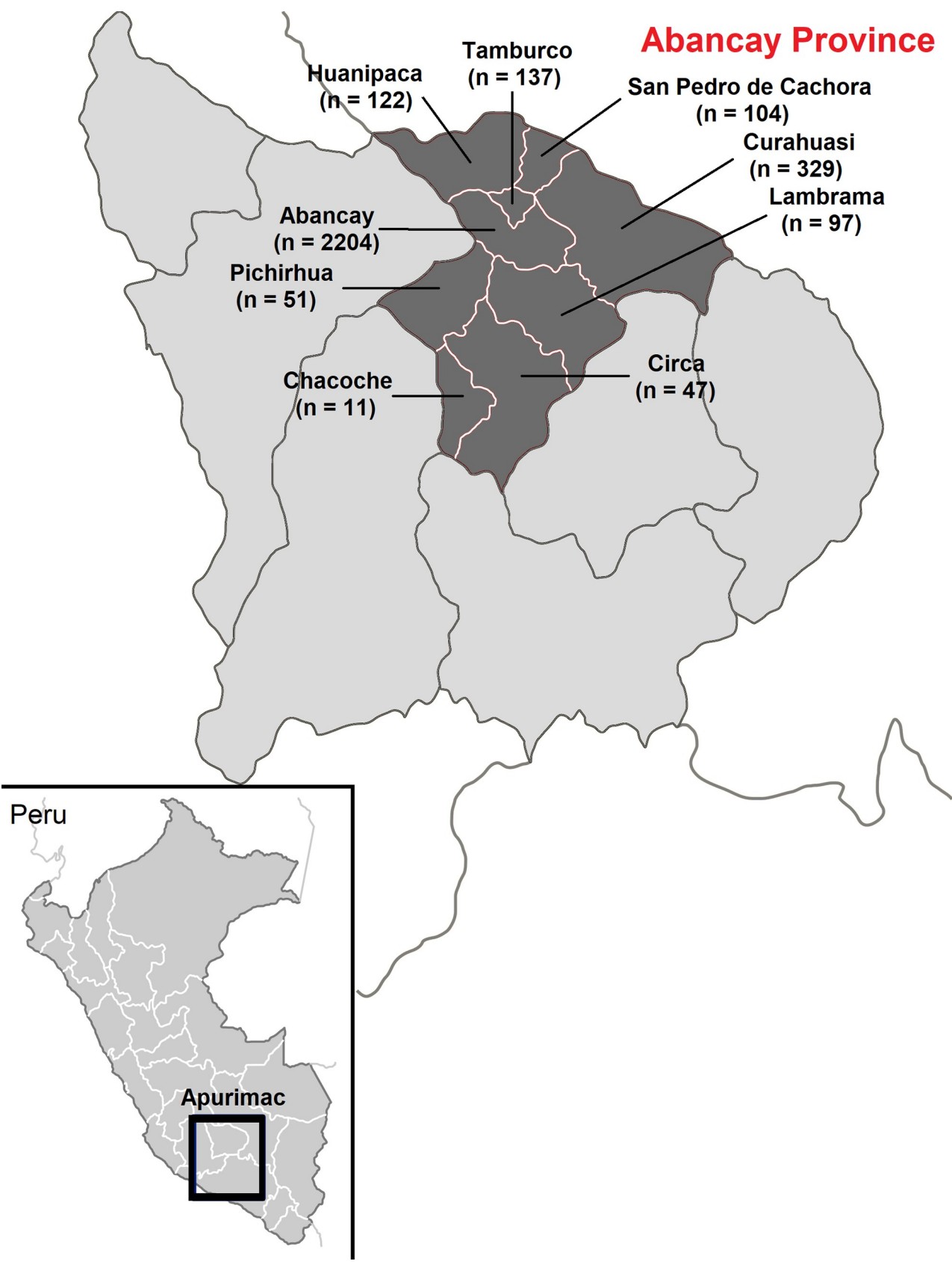

**Fig 1. Geographic location of Abancay province in Peru.**

participants were allocated to the next selected dwelling. If more than one member of the same family shared the same birth date, only one of these members was randomly selected. Men and women of all ages residing for more than 6 months in Abancay were included in the study. People with mental or physical disabilities who had communication difficulties, verbal and written, were excluded from the study. Due to geographical barriers and accessibility in Abancay province, only a total of 3165 participants from 0 to 94 years were included in this study (Fig 1).

## Laboratory analysis

A total volume of 4 ml of venous blood sample was obtained from each participant aged ≤10 years, and 7 ml from each aged >10 years. Serological screening for HBsAg, total anti-HBc and anti-HBs was performed using enzyme-linked immunosorbent assay (ELISA) (Beijing Wantai Biological Pharmacy, Beijing, China) at the Laboratorio de Referencia Nacional Hepatitis, Centro Nacional de Salud Puública of the Instituto Nacional de Salud, Peru. In cases of reactive results for HBsAg, the assay was repeated at least twice. HBsAg was considered confirmed when associated with reactive anti-HBc. If an HBsAg-reactive sample was total anti-HBc negative, it was considered unconfirmed. HBsAg-reactive samples were tested for anti-HBc immunoglobulin M (IgM), hepatitis B e antigen (HBeAg), hepatitis B e antibody (anti-HBe) and anti-HDV, using ELISA (Beijing Wantai Biological Pharmacy). Levels of anti-HBs of ≥10 and <10 mUI/ml were considered as protective and non-protective against HBV infection, respectively, following the manufacturer's instructions (100% sensibility and 99.58% specificity). The HBV viral load was quantified by real-time polymerase chain reaction (PCR), using COBAS AmpliPrep /COBAS TaqMan VHB Test version 2.0 (Roche Molecular Diagnostics, Branchburg, NJ, USA).

## Ethics

The study protocol was approved by the Ethical Committee of the Instituto Nacional de Salud, Lima -Peru. Written informed consent was obtained from all participants aged ≥18 years. For participants aged <18 years, written consent was obtained from their parents or guardians. A translator was available for those participants who were Quechua speakers, who then gave their signed consent to participate in the study.

## Statistical analysis

The prevalence rates of HBsAg, total anti-HBc, and anti-HBs ≥10 mUI/ml (by gender, age and district) with 95% CIs were calculated. The prevalence rate at protective levels of anti-HBs (≥10 mUI/ml) was calculated in participants with HBsAg and total anti-HBc negative. The prevalence rate of anti-HDV was calculated in HBsAg carriers. The prevalence rates (by gender, age and district) were compared using Pearson's $\chi^2$ test. $P$-values of $< 0.05$ were considered statistically significant. All statistical analyses were performed using STATA 16 for Windows (STATA Corporation, College Station, TX, US).

## Results

The overall prevalence rates of HBsAg and total anti-HBc were 1.20% (95% CI 0.85–1.64%) and 41.67% (95% CI 39.95–43.41%), respectively (Table 1). The prevalence rates of HBsAg and anti-HBc total by gender, age and district are shown in Table 1. The prevalence rates of HBsAg and total anti-HBc total in children aged 0–10 years were 0 and 14.40% (95% CI 8.76–21.80%), respectively. Only two participants, aged 15 and 16 years, respectively, were HBsAg positive;

**Table 1. Prevalence rates of HBsAg and total anti-HBc in Abancay, Peru, 2014.**

| | N (%) | HBsAg | | | Total anti-HBc | | |
|---|---|---|---|---|---|---|---|
| | | Positive (n) | Prevalence % (95% CI) | $p^*$ | Positive (n) | Prevalence % (95% CI) | $p^*$ |
| **Overall** | 3165 (100%) | 38 | 1.20% (0.85–1.64) | | 1319 | 41.67% (39.95–43.41) | |
| **Gender** | | | | | | | |
| Male | 2004 (63.3%) | 20 | 0.99% (0.61–1.53) | 0.169 | 829 | 41.36% (39.20–43.55) | 0.645 |
| Female | 1161 (16.7%) | 18 | 1.55% (0.92–2.43) | | 490 | 42.20% (39.34–45.10) | |
| **Age (years)\*\*** | | | | | | | |
| 0–10 | 125 (4.02%) | 0 | 0.0% (0.0–0.0) | 0.657 | 18 | 14.40% (8.76–21.80) | 0.0001 |
| 11–18 | 370 (11.87%) | 3 | 0.81% (0.16–2.35) | | 67 | 18.10% (14.31–22.41) | |
| 19–29 | 933 (30.06%) | 12 | 1.28% (0.66–2.23) | | 245 | 26.25% (23.46–29.20) | |
| 30–59 | 1409 (45.34%) | 19 | 1.34% (0.81–2.09) | | 760 | 53.93% (51.29–56.56) | |
| ≥60 | 270 (8.69%) | 4 | 1.48% (0.40–3.74) | | 203 | 75.18% (69.58–80.22) | |
| **District** | | | | | | | |
| Abancay | 2204 (69.9%) | 26 | 1.18% (0.77–1.72) | 0.092 | 909 | 41.24% (39.17–43.33) | 0.001 |
| Curahuasi | 329 (12.4%) | 7 | 2.12% (0.85–4.33) | | 201 | 61.09% (55.59–66.39) | |
| Tamburco | 137 (4.3%) | 1 | 0.72% (0.02–3.99) | | 47 | 34.30% (26.41–42.89) | |
| Huanipaca | 122 (3.9%) | 0 | 0.0% (0.0–0.0) | | 50 | 40.98% (32.16–50.25) | |
| Lambrana | 97 (3.1%) | 0 | 0.0% (0.0–0.0) | | 34 | 35.05% (25.63–45.40) | |
| San Pedro de Cachora | 104 (3.3%) | 1 | 0.96% (0.02–5.24) | | 34 | 32.69% (23.81–42.58) | |
| Pichirhua | 51 (1.6%) | 2 | 3.92% (0.47–13.45) | | 25 | 49.01% (34.75–63.40) | |
| Circa | 47 (1.5%) | 0 | 0.0% (0.0–0.0) | | 14 | 29.78% (17.33–44.89) | |
| Chacoche | 11 (0.3%) | 1 | 9.09% (0.22–41.27) | | 5 | 45.45% (16.74–76.62) | |

$^*\chi^2$ test.

\*\*Age was obtained in 3107 participants.

therefore, the prevalence rate of HBsAg in children aged <15 years was nil. The prevalence rates of HBsAg among those aged ≥11 years were similar across all age groups (0.81–1.48%), while prevalence rates of total anti-HBc increased significantly with age ($p < 0.0001$) (Table 1). The prevalence rate of anti-HDV was 5.26% (95% CI 0.64–17.74) (2 of the 38 HBsAg carriers, including only participants aged >30 years).

A total of 3135 of 3165 was tested for anti-HBs. Of these, 1201 and 1846 individuals were anti-HBc-positive and anti-HBc-negative, respectively. On the other hand, 36 individuals total anti-HBc-positive had levels of anti-HBs <10 mUI/ml. The prevalence rate at protective levels of anti-HBs (i.e., ≥10 mIU/ml) in individuals who were HBsAg and anti-HBc negative was 66.36% (95% CI 64.15–68.51%) (Table 2). The prevalence rates of anti-HBs (≥10 mUI/ml) in individuals who HBsAg and total anti-HBc negative by gender, age and district are shown in Table 2. The prevalence rate of anti-HBs (≥10 mUI/ml) in children aged 0–10 years was 75.70% (95% CI 66.45–83.47), and the prevalence rates of anti-HBs decreased significantly from the age of 19 years ($p < 0.0001$) (Table 2).

All HBsAg-positive participants ($n = 38$) were HBeAg negative and anti-HBe positive, and only one 30-year-old male participant was anti-HBc IgM positive. HBV viral load was detected in 30 of 38 carriers. Of these 30 carriers, 9 (30%), 15 (50%) and 3 carriers (10%) had HBV DNA levels of <20, 20–1000 and 1000–2000 IU/ml, respectively, and only 3 carriers (10%) had HBV DNA levels of >2000 IU/ml. In a participant with anti-HBc IgM positive the HBV viral load was >2000 IU/ml. In the participants with anti-HDV-positive the HBV viral load were <2000 IU/ml.

**Table 2. Prevalence rate of anti-HBs ≥10 mUI/ml in individuals HBsAg and total anti-HBc negative in Abancay, Peru, 2014.**

|  | N (%) | Positive (n) | Prevalence % (95% CI) | $p^*$ |
|---|---|---|---|---|
| **Overall** | 1846 | 1225 | 66.36% (64.15–68.51) |  |
| **Gender** |  |  |  |  |
| Male | 671 (72.70) | 454 | 67.66% (63.97–71.18 | 0.371 |
| Female | 1175 (27.30 | 771 | 65.61% (62.82–68.33) |  |
| **Age (years)**\*\* |  |  |  |  |
| 0–10 | 107 (5.90) | 81 | 75.70% (66.45–83.47) | 0.0001 |
| 11–18 | 303 (16.70) | 281 | 92.73% (89.21–95.39) |  |
| 19–29 | 688 (37.93) | 522 | 75.87% (72.49–79.02) |  |
| 30–59 | 649 (35.78) | 307 | 47.30% (43.40–51.22) |  |
| ≥60 | 67 (3.69) | 18 | 26.86% (16.76–39.09) |  |
| **District** |  |  |  |  |
| Abancay | 1295 (70.15) | 872 | 67.33% (64.70–69.88) | 0.0001 |
| Curahuasi | 191 (10.35) | 114 | 59.68% (52.36–66.70) |  |
| Tamburco | 90 (4.88) | 62 | 68.88% (58.26–78.23) |  |
| Huanipaca | 72 (3.90) | 42 | 58.33% (46.11–69.84) |  |
| San Pedro de Cachora | 70 (3.79) | 36 | 51.42% (39.17–63.55) |  |
| Lambrama | 63 (3.41) | 51 | 80.95% (69.09–89.75) |  |
| Circa | 33 (1.79) | 21 | 63.63% (45.12–79.60) |  |
| Pichirhua | 26 (1.41) | 25 | 96.15% (80.36–99.90) |  |
| Chacoche | 6 (0.33) | 2 | 33.33% (4.32–77.72) |  |

$^*\chi^2$ tests.

\*\*Age was obtained in 1814 participants.

## Discussion

Our findings showed that since the introduction of the first pilot vaccination program against HBV in Abancay, there has been a decrease in the HBsAg carrier rate over the past two decades, from high to low endemicity (9.8% in 1991 vs. 1.2% in 2014), compared with those reported in previous studies, especially in children (Fig 2) [11]. Our study did not find chronic HBsAg carriers among children aged <15 years, whereas with other studies which showed reduced HBsAg carrier rates from 9.8% to 2.3% in children aged 5–14 years and 1% in those aged <1 year in China and 0.5% in children aged <11 years in Colombia [6,15]. Therefore, our study showed that the vaccination program is already preventing new chronic infections in Abancay. In addition, infant vaccination is reducing mortality from HBV-related liver diseases, including cirrhosis, hepatocarcinoma and fulminant hepatitis, as reported previously [14].

In this study, we also observed a change in the prevalence of HDV among HBsAg carriers, from 9% in 1990 [11] to 5.2% in 2014, 23 years since the launch of the vaccination program, and HBV infection was not detected in those aged <30 years, thus correlating with the reduced HBsAg carrier rate. These findings are in agreement with other studies, which reported a decrease in the prevalence of HDV in chronic HBsAg carriers, following vaccination against HBV [19]. Moreover, several studies suggested that high vaccination coverage against HBV can eliminate both HBV and HDV infections [20]. In our study most HBsAg carriers had HBV viral load low (<2000 IU/ml). HBV DNA level is closely associated with stadium to the infection. Therefore, it likely these to have been infected in adulthood, and it are experiencing

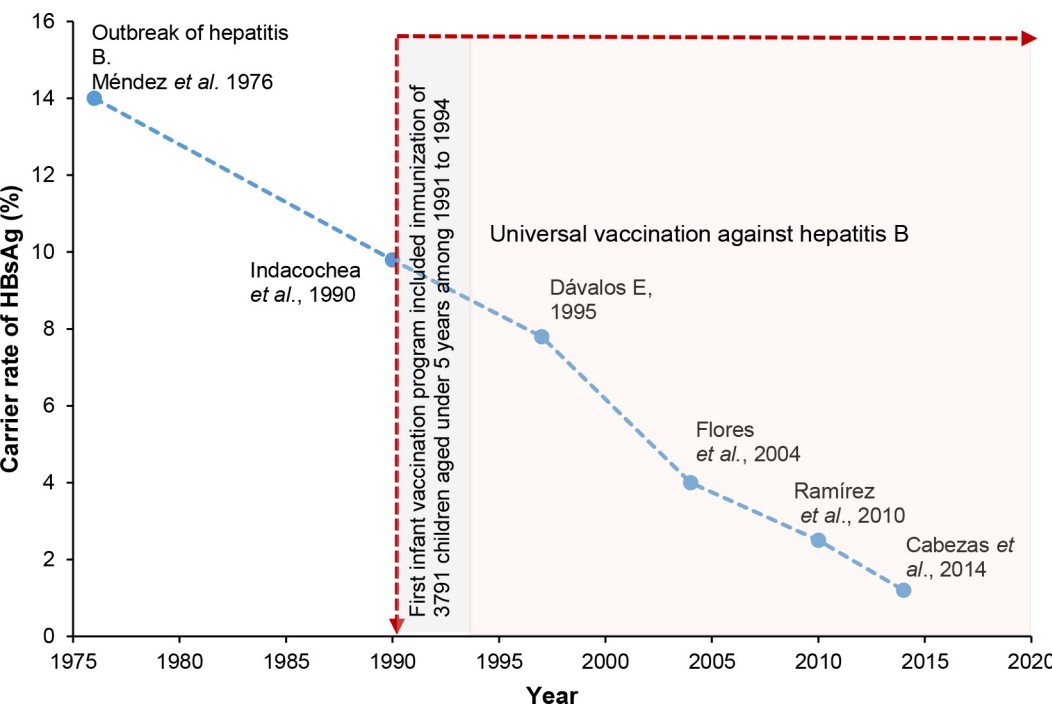

**Fig 2. Carrier rates of HBsAg, compared with HBsAg carrier rates reported from previous studies, in Abancay province in Peru [11,12,16–18].**

viral clearance as the natural course of the infection progressed. However, in this study we did not follow to HBsAg carriers.

Anti-HBc-positive/HBsAg-negative indicates previous HBV infection [21]. In this study, our findings demonstrated a high anti-HBc prevalence rate in Abancay province, and this prevalence rate was increased significantly with age. One possible reason for this is that those aged >18 years had greater exposure to HBV before the introduction of the vaccination program; hence, the probability of anti-HBc positivity increased in individuals aged >18 years. Moreover, there were two outbreaks of HBV infection in Abancay [16] before the introduction of the vaccination program, which would also contribute to high anti-HBc prevalence rates in those aged ≥18 years. Therefore, the increasing prevalence of total anti-HBc with age suggested natural immunity. Moreover, the anti-HBc prevalence was lower in children and adolescents. These findings also confirm that the prevalence rate of anti-HBc has declined since the introduction of the vaccination program in Abancay province. These findings are consistent with other studies, which found low total anti-HBc carrier rates in children, compared with those aged ≥18 years where seropositivity of total anti-HBc increased with the age [22].

In our study, we found high prevalence rates of anti-HBs at protective levels (i.e., ≥10 mIU/ml) in individuals aged 0–18 years who were HBsAg and anti-HBc negative. These high prevalence rates of anti-HBs in individuals aged 0–18 years show that HBV vaccination program has been successfully implemented for >20 years with good coverage and efficacy among the children in Abancay, Peru. Levels of anti-HBs considered protective against HBV are an indicator of successful vaccination. However, in this study, low prevalence rates of anti-HBs at protective levels were obtained in individuals aged 30–59 years who were HBsAg and anti-HBc negative, and reached its lowest level in those aged ≥60 years. This is probably related to the fact that these individuals either did not have access to vaccination, because of a low coverage of hepatitis B vaccination in the young/older adult group, or did not respond to the vaccine,

possibly due to either a defense mechanism or changes in their immune response. However, our study methodology here did not allow for us to confirm this hypothesis. This observation was previously described by other researchers such as Zaffina *et al.* [23], who showed that repeated vaccinations in this group of people did not alter the levels of memory B cells or the production of anti-HBs antibodies. Therefore, further research is necessary, especially on host immune response, in the context of implementation of HBV elimination programs, with the aim to, at least, reduce to a minimum the number of people susceptible to infection, if not to eliminate HBV.

The main limitation of this study relates to different time periods (1990 vs. 2014) used in comparing HBsAg carrier rates in different populations, as a result of the HBV vaccination program, since there were no cohort effects in children aged under 5 years who were vaccinated between 1991 and 1994. However, the effects of the HBV vaccination program that was introduced in 1991 would manifest in an age-specific manner and would be expected mainly in individuals aged <30 years, as shown by Chang *et al.* [9] who compared data from 1986 and 1994 [9]. Another study limitation is that the number of participants included in this study was smaller than the required sample size, and the proportion of women was higher than that of men, in accordance with other studies [24,25]. Given these limitations, it is possible that carrier rates were either underestimated or overestimated, thus resulting in potential bias. Finally, the prevalence rates of HBV varies from distric to distric, particularly as some district have small population, therefore, it can lead to a bias. However, according to used methods, the objective of this study was to determine the prevalence of HBV and HDV infections in Abancay province. Despite these limitations, this population-based study showed that vaccination against HBV has a positive impact by reducing HBsAg carrier rates in the general population and in children aged <15 years in Abancay, Peru.

## Conclusions

Our study findings showed that HBV prevalence has changed from high to low endemicity, 23 years after the introduction of the vaccination program against HBV in Abancay. It is noteworthy that there are no chronic HBsAg carriers among children aged <15 years and no HDV infection has been detected in those aged under 30 years. Moreover, we found high prevalence rates of anti-HBs at protective levels in individuals aged 0–18 years, and low prevalence rates of anti-HBs at protective levels in individuals aged older 29 years who were HBsAg and anti-HBc negative. These findings highlight the necessity to strengthen the vaccination program against HBV in those aged older 29 years, as well as to diagnose and treat in a timely manner chronic HBsAg carriers, with a view to eliminating HBV in the future.

## Supporting information

**S1 Data.**
(XLS)

## Acknowledgments

This study was funded by the Instituto Nacional de Salud, Peru. We thank to the researchers and professionals at Centro Nacional de Salud Puública, Instituto Nacional de Salud, Lima, Peru, DIRESA Apurímac and Ministerio de Salud (MINSA).

## Author Contributions

**Conceptualization:** Cesar Cabezas, Omar Trujillo, Johanna Balbuena, Flor de Maria Peceros, Magna Suárez.

**Data curation:** Cesar Cabezas, Omar Trujillo, Johanna Balbuena, Flor de Maria Peceros, Manuel Terrazas, Magna Suárez, Luis Marin, Max Carlos Ramírez-Soto.

**Formal analysis:** Cesar Cabezas, Omar Trujillo, Johanna Balbuena, Max Carlos Ramírez-Soto.

**Funding acquisition:** Cesar Cabezas, Omar Trujillo, Johanna Balbuena, Flor de Maria Peceros, Manuel Terrazas, Magna Suárez, Luis Marin, Max Carlos Ramírez-Soto.

**Investigation:** Cesar Cabezas, Omar Trujillo, Johanna Balbuena, Flor de Maria Peceros, Manuel Terrazas, Magna Suárez, Luis Marin, Janet Apac.

**Methodology:** Cesar Cabezas, Omar Trujillo, Flor de Maria Peceros, Manuel Terrazas, Magna Suárez, Luis Marin, Janet Apac, Max Carlos Ramírez-Soto.

**Project administration:** Cesar Cabezas, Johanna Balbuena, Manuel Terrazas, Luis Marin.

**Resources:** Flor de Maria Peceros.

**Software:** Max Carlos Ramírez-Soto.

**Supervision:** Cesar Cabezas, Omar Trujillo.

**Validation:** Cesar Cabezas.

**Visualization:** Manuel Terrazas.

**Writing – original draft:** Cesar Cabezas, Omar Trujillo, Johanna Balbuena, Flor de Maria Peceros, Manuel Terrazas, Magna Suárez, Luis Marin, Janet Apac, Max Carlos Ramírez-Soto.

**Writing – review & editing:** Cesar Cabezas, Omar Trujillo, Johanna Balbuena, Flor de Maria Peceros, Manuel Terrazas, Magna Suárez, Luis Marin, Janet Apac, Max Carlos Ramírez-Soto.

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
