## [Decision Letter · Decision Letter 0]

11 Feb 2020

PONE-D-19-35864

Decrease in the prevalence of hepatitis B and D virus infections in an endemic area in Peru 23 years after the introduction of the first pilot vaccination program against hepatitis B

PLOS ONE

Dear Dr. Ramírez-Soto,

Thank you for submitting your manuscript to PLOS ONE. After careful consideration, we feel that it has merit but does not fully meet PLOS ONE’s publication criteria as it currently stands. Therefore, we invite you to submit a revised version of the manuscript that addresses the points raised during the review process.

Your manuscript was reviewed by 3 experts in the field. All reviewers were very critical of your submission and produced many important comments. Please carefully review the attached comments and provide thorough responses to each point.

We would appreciate receiving your revised manuscript by Mar 27 2020 11:59PM. To enhance the reproducibility of your results, we recommend that if applicable you deposit your laboratory protocols in protocols.io, where a protocol can be assigned its own identifier (DOI) such that it can be cited independently in the future. For instructions see: http://journals.plos.org/plosone/s/submission-guidelines#loc-laboratory-protocols

We look forward to receiving your revised manuscript.

Kind regards,

Yury E Khudyakov, PhD

Academic Editor

PLOS ONE

Journal Requirements:

Reviewers' comments:

Reviewer's Responses to Questions

**Comments to the Author**

1. Is the manuscript technically sound, and do the data support the conclusions?

Reviewer #1: Yes

Reviewer #2: Partly

Reviewer #3: Partly

2. Has the statistical analysis been performed appropriately and rigorously? 

Reviewer #1: Yes

Reviewer #2: No

Reviewer #3: Yes

3. Have the authors made all data underlying the findings in their manuscript fully available?

Reviewer #1: Yes

Reviewer #2: No

Reviewer #3: Yes

4. Is the manuscript presented in an intelligible fashion and written in standard English?

Reviewer #1: Yes

Reviewer #2: No

Reviewer #3: Yes

5. Review Comments to the Author

Reviewer #1: The study performed by Cabezas and colleagues demonstrated HBV prevalence reduction 23 years after the introduction of a pilot vaccination program in the high prevalence area of Abancay, Peru. The study has merit, however, some points should be addressed before publication.

Keywords (line 40): It would be more informative to add Peru as a keyword instead Abancay.

Material and Methods:

The number and exact location of study population is not clear. How many people from each district/conglomerate were selected? It is partially set in table 1, but should be better explained. How many individuals are from urban areas and how many are from rural areas? Please, clarify these points. I suggest this information to be added to Figure 1 to make it more informative. Figure 1 could contain more details about the study population (number of subjects/district and district locations). As it stands, it is very similar to other figures available on various websites (wikipedia etc).

Line 91: What the authors mean as “each geographical area”? It should be better explained. I suggest it to be placed in figure 1.

Line 92: The authors mean “A total of 113 conglomerates comprising 32 inhabitants each ? If yes, please add this word.

Lines 95-95: In the sentence “In cases of rejections or losses, the houses were not replaced by the next eligible house on the list”, is the word ‘not’ placed correctly? If yes, the sentence is not relevant. If no, please make the correction.

Results:

Line 133: How many individuals were recruited and what was the loss? Please, provide this information.

Figure 2: Please, rephrase the caption. It is not clear, the text could be improved. Also, on the x-axis of the chart, the term “Carrier rate of HBsAg (%)” is confusing. Please replace to "HBsAg carriers (%)" or just "HBsAg (%)".

Lines 135-137: The sentence “A decreased trend in the carrier rates of HBsAg, compared with those reported in previous studies, in populations of Abancay 24 years since the introduction of the first pilot vaccination program against HBV was observed (9.8% in 1991 vs. 1.2% in 2014 (Fig 2) [11].” is more a discussion than a result. Also, in other points of the paper, the authors set 23 instead 24 years since the first vaccination program. Please, standardize the time interval.

Line 143: How was the viral load and serological profile of the HBV acute infected subject?

Lines 146-147: Did the anti-HDV carriers present higher HBV viral loads?

Lines 148-154: I suggest to replace “carrier rates of HBsAg, anti-HBc and anti-HBs” by “HBsAg anti-HBc and anti-HBs prevalences” or “HBsAg anti-HBc and anti-HBs positivity”.

Discussion:

Lines 169-170: “In this study, we also observed a decline in the prevalence of HDV among HBsAg carriers, from 9% in 1990 [11] to 5.2% in 2014”. Was this decrease statistically significative?

Lines 186-190: This is probably related to the fact that these individuals either did not have access to vaccination or did not respond to the vaccine.One possible explanation is that those with repeated HBV exposure no longer became infected and hence did not mount protective levels of anti-HBs, possibly due to either a defense mechanism or changes in their immune response”.If I understood correctly, in this sentence the authors attributed the anti-HBs levels (that might be due to low vaccination response) to repeated HBV exposures over time. It is a quite speculative and unlikely to happen, once individuals exposed to HBV usually have positivity for both anti-HBs and anti-HBc serological markers. Please clarify the sentence.

Other comments:

-Is there any difference in HBV prevalences between urban and rural areas?

-Based on table one, individuals from nine districts have been recruited, most of them to Abancay, where the highest prevalences of HBV serological markers were observed. It can lead to a bias and should be better discussed.

-The age groups showed in table 1 could be standardized and better explored/discussed

-The authors performed molecular tests (HBV viral loads) but did not provide any discussion exploring these points.

Reviewer #2: This study presented the change of hepatitis B and D virus infection in Peru 23 years after the implementation of HBV vaccination, but this is not novel as previous study has already showed such pattern [12-14]. Although the authors mentioned that there are no data on the impact of the HBV vaccination program on the carrier rates of HBsAg, anti-HBc, anti-HBs and anti-HDV. Apart from the change of HBsAg in Figure, there was no data to support the changes of other antigens/antibodies. In addition, the step of sample collection is not clear to me. For instance, what is the meaning of conglomerates? Why 'in case of rejection or losses, the houses were not replaced by the next eligible house on the list'? What's more, the statistical analysis is not appropriate. To my knowledge, there is no 'by bivariate analysis, using chi-square test and odds ratios'. Odds ratios were obtained via logistic regression. And it seems that Table used such analysis but it could be done by using simple chi-square test. The detail of my comments are as below:

1. The description seems not consistent:

Line 88-89 Study population and sample method: the author calculated the sample size should be at least 3520, but only 3165 were included in this study.

2. Line 99-100: why people with mental or physical disabilities...were excluded from this study?

3. Line 112-113: Levels of anti-HBs of >=10 and <10mUI/ml were considered as protective and non-protective...How did the author get the cutoff? Any reference to support this? What's the sensitivity and specificity?

4. Line 125: how did the authors calculate the 95% CIs?

5. Table 2: apart from age and gender, should not the paper has other variables? It would be interesting to check the other characteristics of antigen prevalence?

6. Line 162: the reduction of hbv prevalence was purely derived from the comparison with previous study in the region. Therefore, it is very important to show the similarity between the previous and the current study. For instance, the mean age, gender distribution?

7. Line 166-168: Therefore, our study showed that...reducing mortality...This study did not provide any data about the liver cancer mortality?

8. Line 169: we also observed a decline in the prevalence of HDV...No data from the result section was presented for the prevalence of HDV.

9. Line 176-183: this paragraph is very confusing and I could see the logic here.

Reviewer #3: The authors conducted a survey on hepatitis B and D infection in Peru after a Hepatitis B vaccination program started. The results of the survey were compared to survey results from the times before the vaccination regulations. The prevalence of chronic hepatitis B declined and surprisingly the prevalence of anti-HBc was higher than the prevalence of anti-HBs. Anti-HBs positive, anti-HBc negative results indicate a vaccination. It was not shown how many participants had this constellation. Anti-HBc positive indicates an infection with HBV. About 40 % of the population had a HBV infection. This is an astonishing result for a population that had been vaccinated.

Major concerns

1) The authors conceive the comparison of two surveys conducted many years apart as a study. I would say the survey in 2014 was a study and the results of this study should be compared to the previous publication in the results section.

2) Are the results of the survey conducted in 2014 published elsewhere? It is surprising that it took six years to present the results. Please comment.

3) Why do the findings highlight the necessity to strengthen the childhood vaccination program? It is not known whether vaccination was performed or not. Please explain.

Minor concerns

1) Discussion line 161: continuous decrease? You compare two different survey and you have now information on what happened between them. So delete continuous.

2) line 163: in contrast? Why in contrast?

6. PLOS authors have the option to publish the peer review history of their article (what does this mean?). If published, this will include your full peer review and any attached files.

Reviewer #1: No

Reviewer #2: No

Reviewer #3: Yes: Albert Nienhaus

---

## [Author Response · Author response to Decision Letter 0]

13 May 2020

Response-to-reviewers: Manuscript PONE-D-19-35864

We thank the Reviewers for their comments and constructive criticism, we believe that the quality of our manuscript has been significantly improved. We have revised our paper in a point-by-point manner. Modifications are in yellow text. 

Reviewer #1: The study performed by Cabezas and colleagues demonstrated HBV prevalence reduction 23 years after the introduction of a pilot vaccination program in the high prevalence area of Abancay, Peru. The study has merit, however, some points should be addressed before publication.

Keywords (line 40): It would be more informative to add Peru as a keyword instead Abancay. 

Response: Thank you for this suggestion; we have corrected the keywords (Line 40).

Material and Methods:

The number and exact location of study population is not clear. How many people from each district/conglomerate were selected? It is partially set in table 1, but should be better explained. How many individuals are from urban areas and how many are from rural areas? Please, clarify these points. I suggest this information to be added to Figure 1 to make it more informative. Figure 1 could contain more details about the study population (number of subjects/district and district locations). As it stands, it is very similar to other figures available on various websites (wikipedia etc). 

Response: Thank you for this suggestion; we have corrected the Fig. 1.

Line 91: What the authors mean as “each geographical area”? It should be better explained. I suggest it to be placed in figure 1. 

Response: Thank you for this suggestion; we have corrected the Fig. 1.

Line 92: The authors mean “A total of 113 conglomerates comprising 32 inhabitants each ? If yes, please add this word. 

Response: Thank you for this suggestion; we have corrected the text (line 93).

Lines 95-95: In the sentence “In cases of rejections or losses, the houses were not replaced by the next eligible house on the list”, is the word ‘not’ placed correctly? If yes, the sentence is not relevant. If no, please make the correction. 

Response: Thank you for this suggestion; we have corrected the sentence (line 95-96). 

Results:

Line 133: How many individuals were recruited and what was the loss? Please, provide this information. 

Response: Thank you for this suggestion; due to geographical barriers and accessibility in Abancay province, only a total of 3165 participants were included in this study (lenes 101-102).

Figure 2: Please, rephrase the caption. It is not clear, the text could be improved. Also, on the x-axis of the chart, the term “Carrier rate of HBsAg (%)” is confusing. Please replace to "HBsAg carriers (%)" or just "HBsAg (%)". 

Response: Thank you for this suggestion; we have corrected the Fig. 2.

Lines 135-137: The sentence “A decreased trend in the carrier rates of HBsAg, compared with those reported in previous studies, in populations of Abancay 24 years since the introduction of the first pilot vaccination program against HBV was observed (9.8% in 1991 vs. 1.2% in 2014 (Fig 2) [11].” is more a discussion than a result. Also, in other points of the paper, the authors set 23 instead 24 years since the first vaccination program. Please, standardize the time interval. 

Response: Thank you for this suggestion; the paragraph has been included in the Discussion section (lines 161-164)

Line 143: How was the viral load and serological profile of the HBV acute infected subject?

Response: Thank you for this suggestion; In a participant with anti-HBc IgM positive the HBV viral load was >2000 IU/ml (line 151).

Lines 146-147: Did the anti-HDV carriers present higher HBV viral loads? 

Reponse: Thank you for this suggestion; In the participants with anti-HDV positive the HBV viral load were <2000 IU/ml (lines 151-152).

Lines 148-154: I suggest to replace “carrier rates of HBsAg, anti-HBc and anti-HBs” by “HBsAg anti-HBc and anti-HBs prevalences” or “HBsAg anti-HBc and anti-HBs positivity”. 

Response: Thank you for this suggestion; we have corrected these typos (lines 135-137 and 140-146).

Discussion:

Lines 169-170: “In this study, we also observed a decline in the prevalence of HDV among HBsAg carriers, from 9% in 1990 [11] to 5.2% in 2014”. Was this decrease statistically significative? 

Response: Thank you for this suggestion; a statistical comparison cannot be performed. We have corrected this paragraph (line 175).

Lines 186-190: This is probably related to the fact that these individuals either did not have access to vaccination or did not respond to the vaccine.One possible explanation is that those with repeated HBV exposure no longer became infected and hence did not mount protective levels of anti-HBs, possibly due to either a defense mechanism or changes in their immune response”.If I understood correctly, in this sentence the authors attributed the anti-HBs levels (that might be due to low vaccination response) to repeated HBV exposures over time. It is a quite speculative and unlikely to happen, once individuals exposed to HBV usually have positivity for both anti-HBs and anti-HBc serological markers. Please clarify the sentence.

Response: Thank you for this suggestion; we have corrected this paragraph (Lines 196-199). 

Other comments:

Is there any difference in HBV prevalences between urban and rural areas? 

Response: Thank you for this suggestion; difference in HBV prevalences between urban and rural areas was not an objective of the study. These data were not available.

Based on table one, individuals from nine districts have been recruited, most of them to Abancay, where the highest prevalences of HBV serological markers were observed. It can lead to a bias and should be better discussed. 

Response: Thank you for this suggestion. We have included a limitation (lines 214-217). 

The age groups showed in table 1 could be standardized and better explored/discussed. 

Response: Thank you for this suggestion; The age groups was standardized in this way to be comparable with other studies.

The authors performed molecular tests (HBV viral loads) but did not provide any discussion exploring these points. 

Response: Thank you for this suggestion; we have included a paragraph on HBV viral load (181-184).

Reviewer #2: This study presented the change of hepatitis B and D virus infection in Peru 23 years after the implementation of HBV vaccination, but this is not novel as previous study has already showed such pattern [12-14]. Although the authors mentioned that there are no data on the impact of the HBV vaccination program on the carrier rates of HBsAg, anti-HBc, anti-HBs and anti-HDV. 

Apart from the change of HBsAg in Figure, there was no data to support the changes of other antigens/antibodies. In addition, the step of sample collection is not clear to me. 

Response: Thank you for your comment. In the previous study there were no data on anti-HBc and anti-HBs.

For instance, what is the meaning of conglomerates? 

Response: Thank you for your comment. Conglomerate are population groups (line 91). 

Why 'in case of rejection or losses, the houses were not replaced by the next eligible house on the list'? 

Response: Thank you for your comment. We have corrected this tipo (line 95-96).

What's more, the statistical analysis is not appropriate. To my knowledge, there is no 'by bivariate analysis, using chi-square test and odds ratios'. Odds ratios were obtained via logistic regression. And it seems that Table used such analysis but it could be done by using simple chi-square test. 

Response: Thank you for your comment. Statistical analyzes were reviewed with our statistician (lines 129-130 and Table 2).

1. The description seems not consistent:

Line 88-89 Study population and sample method: the author calculated the sample size should be at least 3520, but only 3165 were included in this study. 

Response: Thank you for this suggestion; due to geographical barriers and accessibility in Abancay province, only a total of 3165 participants were included in this study (lines 1011-102).

2. Line 99-100: why people with mental or physical disabilities...were excluded from this study? 

Response: Thank you for your comment. We didn't have informed consent from them.

3. Line 112-113: Levels of anti-HBs of >=10 and <10mUI/ml were considered as protective and non-protective...How did the author get the cutoff? Any reference to support this? What's the sensitivity and specificity? 

Response: Thank you for your comment. Levels of anti-HBs of �10 and <10 mUI/ml were considered as protective and non-protective against HBV infection, respectively, following the manufacturer’s instructions (Beijing Wantai Biological Pharmacy, Beijing, China) (lines 114-115).

4. Line 125: how did the authors calculate the 95% CIs? 

Response: Thank you for your comment. 95% CIs were calculated with the Statistical Program. 

5. Table 2: apart from age and gender, should not the paper has other variables? It would be interesting to check the other characteristics of antigen prevalence? 

Response: Thank you for your comment. For the purposes of this study, prevalence by district was included (Table 1).

6. Line 162: the reduction of hbv prevalence was purely derived from the comparison with previous study in the region. Therefore, it is very important to show the similarity between the previous and the current study. For instance, the mean age, gender distribution? 

Response: Thank you for this suggestion; a statistical comparison cannot be performed. We have corrected this paragraph (lines 161-164).

7. Line 166-168: Therefore, our study showed that...reducing mortality...This study did not provide any data about the liver cancer mortality? 

Response: Thank you for this suggestion. We have corrected this paragraph (168-170).

8. Line 169: we also observed a decline in the prevalence of HDV...No data from the result section was presented for the prevalence of HDV. 

Response: Thank you for this suggestion. We have corrected this paragraph. The HDV prevalence are shown in the Results section (lines 137-138).

9. Line 176-183: this paragraph is very confusing and I could see the logic here. 

Response: Thank you for this suggestion. We have corrected this paragraph (186-193).

Reviewer #3: The authors conducted a survey on hepatitis B and D infection in Peru after a Hepatitis B vaccination program started. The results of the survey were compared to survey results from the times before the vaccination regulations. The prevalence of chronic hepatitis B declined and surprisingly the prevalence of anti-HBc was higher than the prevalence of anti-HBs. Anti-HBs positive, anti-HBc negative results indicate a vaccination. It was not shown how many participants had this constellation. Anti-HBc positive indicates an infection with HBV. 

About 40 % of the population had a HBV infection. This is an astonishing result for a population that had been vaccinated.

Response: Thank you for this suggestion. The objective of our study was to determine the prevalence of HBV and HDV infections in Abancay 23 years since the introduction of the first pilot vaccination program against HBV, as well as the post-vaccine response against hepatitis B (lines 67-70).

Major concerns

1) The authors conceive the comparison of two surveys conducted many years apart as a study. I would say the survey in 2014 was a study and the results of this study should be compared to the previous publication in the results section. 

Response: Thank you for this suggestion. The results of both studies are compared in the Discussion section. 

2) Are the results of the survey conducted in 2014 published elsewhere? It is surprising that it took six years to present the results. Please comment. 

Response: Our delay was due to logistical, administrative and financial difficulties in completing the study.

3) Why do the findings highlight the necessity to strengthen the childhood vaccination program? It is not known whether vaccination was performed or not. Please explain. 

Response: Thank you for this suggestion. We have corrected this sentence. These findings highlight the necessity to strengthen the vaccination program against HBV in individuals aged >18 years, as well as to diagnose and treat in a timely manner chronic HBsAg carriers, with a view to eliminating HBV in the future (lines 225-226).

Minor concerns

1) Discussion line 161: continuous decrease? You compare two different survey and you have now information on what happened between them. So delete continuous. 

Response: Thank you for this suggestion. We have corrected this sentence (line 162).

2) line 163: in contrast? Why in contrast? 

Response: Thank you for this suggestion. We have corrected this

---

## [Decision Letter · Decision Letter 1]

1 Jun 2020

PONE-D-19-35864R1

Decrease in the prevalence of hepatitis B and D virus infections in an endemic area in Peru 23 years after the introduction of the first pilot vaccination program against hepatitis B

PLOS ONE

Dear Dr. Ramírez-Soto,

Thank you for submitting your manuscript to PLOS ONE. After careful consideration, we feel that it has merit but does not fully meet PLOS ONE’s publication criteria as it currently stands. Therefore, we invite you to submit a revised version of the manuscript that addresses the points raised during the review process.

Your manuscript was reviewed by 3 original reviewers. Although 2 reviewers were satisfied with your responses, one reviewer still identified some points that require your attention.

We look forward to receiving your revised manuscript.

Kind regards,

Yury E Khudyakov, PhD

Academic Editor

PLOS ONE

Reviewers' comments:

Reviewer's Responses to Questions

**Comments to the Author**

1. If the authors have adequately addressed your comments raised in a previous round of review and you feel that this manuscript is now acceptable for publication, you may indicate that here to bypass the “Comments to the Author” section, enter your conflict of interest statement in the “Confidential to Editor” section, and submit your "Accept" recommendation.

Reviewer #1: All comments have been addressed

Reviewer #2: (No Response)

Reviewer #3: All comments have been addressed

2. Is the manuscript technically sound, and do the data support the conclusions?

Reviewer #1: Yes

Reviewer #2: Partly

Reviewer #3: Yes

3. Has the statistical analysis been performed appropriately and rigorously? 

Reviewer #1: Yes

Reviewer #2: No

Reviewer #3: Yes

4. Have the authors made all data underlying the findings in their manuscript fully available?

Reviewer #1: Yes

Reviewer #2: Yes

Reviewer #3: Yes

5. Is the manuscript presented in an intelligible fashion and written in standard English?

Reviewer #1: Yes

Reviewer #2: Yes

Reviewer #3: Yes

6. Review Comments to the Author

Reviewer #1: (No Response)

Reviewer #2: Thanks for the authors responses to my comments. As responded by the author, the novity of this study was the results about anti-HBc (~42%) and anti-HBs (~39%). Based on the low prevalence of HBsAg, I assume the increasing prevalence of anti-HBc with age suggested natural immunity while the decreasing prevalence of anti-HBs with age correlated with the immunization program. But such kind of summary was not found in the current paper. I agree the reduction of HBsAg was consistent with previous study which showed validity of the data. But I would recommend the author wrote more about the novel results and the interpretation of them.

My other concern is still about the statistical test and presentation of the results. For instance,

1. Table 1. As the study showed the prevalence, there should not be any 95% CI include negative values. But this is the case in a lot of numbers.

2. Table 2. The author used chi-square test, but what the reference group and how large are the samples for each group. As this is not an adjusted analysis, why did not the authors combine Table 2 with Table 1? By doing this, more statistical comparison could also be provided.

Reviewer #3: The comments of the reviewers were adressed and the manuscript improved. It is ready for publication now

7. PLOS authors have the option to publish the peer review history of their article (what does this mean?). If published, this will include your full peer review and any attached files.

Reviewer #1: No

Reviewer #2: Yes: Wen-Qiang He

Reviewer #3: Yes: Albert Nienhaus

---

## [Author Response · Author response to Decision Letter 1]

13 Jul 2020

Response-to-reviewers: Manuscript PONE-D-19-35864R1

We thank the Reviewers for their comments and constructive criticism, we believe that the quality of our manuscript has been significantly improved. We have revised our paper in a point-by-point manner. Modifications are in yellow text. 

Reviewer #2: Thanks for the authors responses to my comments. As responded by the author, the novity of this study was the results about anti-HBc (~42%) and anti-HBs (~39%). Based on the low prevalence of HBsAg, I assume the increasing prevalence of anti-HBc with age suggested natural immunity while the decreasing prevalence of anti-HBs with age correlated with the immunization program. But such kind of summary was not found in the current paper. I agree the reduction of HBsAg was consistent with previous study which showed validity of the data. But I would recommend the author wrote more about the novel results and the interpretation of them.

Response: Thank you for this suggestion; We have corrected these paragraphs in Discussion section (lines 205-214 and lines 218-228).

My other concern is still about the statistical test and presentation of the results. For instance,

1. Table 1. As the study showed the prevalence, there should not be any 95% CI include negative values. But this is the case in a lot of numbers.

Response: Thank you for this suggestion; we have corrected Table 1.

2. Table 2. The author used chi-square test, but what the reference group and how large are the samples for each group. As this is not an adjusted analysis, why did not the authors combine Table 2 with Table 1? By doing this, more statistical comparison could also be provided. 

Response: Thank you for this suggestion; we have combine Table 2 with Table 1 “Prevalence rates of HBsAg and anti-HBc”, include the p-values. For a better understanding of the readers, we have also include a Table 2 “prevalence rates of anti-HBs at protective levels (i.e., �10 mIU/ml) in individuals who were HBsAg and anti-HBc negative”. We have also re-estimated the prevalence rates of anti-HBs (lines 160-167 and Table 2).

---

## [Editor Report · Decision Letter 2]

20 Jul 2020

Decrease in the prevalence of hepatitis B and D virus infections in an endemic area in Peru 23 years after the introduction of the first pilot vaccination program against hepatitis B

PONE-D-19-35864R2

Dear Dr. Ramírez-Soto,

We’re pleased to inform you that your manuscript has been judged scientifically suitable for publication and will be formally accepted for publication once it meets all outstanding technical requirements.

Kind regards,

Yury E Khudyakov, PhD

Academic Editor

PLOS ONE
---

## [Editor Report · Acceptance letter]

27 Jul 2020

PONE-D-19-35864R2 

Decrease in the prevalence of hepatitis B and D virus infections in an endemic area in Peru 23 years after the introduction of the first pilot vaccination program against hepatitis B 

Dear Dr. Ramírez-Soto:

I'm pleased to inform you that your manuscript has been deemed suitable for publication in PLOS ONE. Congratulations! Your manuscript is now with our production department. 

Kind regards, 

on behalf of

Dr. Yury E Khudyakov 

Academic Editor

PLOS ONE